# Zero-1-to-G: Taming Pretrained 2D Diffusion Model for Direct 3D Generation

**Xuyi Meng**\*                                            *mengxuyi@seas.upenn.edu*
*University of Pennsylvania*

**Chen Wang**\*                                            *chenw30@seas.upenn.edu*
*University of Pennsylvania*

**Jiahui Lei**                                            *leijh@seas.upenn.edu*
*University of Pennsylvania*

**Kostas Daniilidis**                                            *kostas@cis.upenn.edu*
*University of Pennsylvania*

**Jiatao Gu**                                            *jgu32@cis.upenn.edu*
*University of Pennsylvania*

**Lingjie Liu**                                            *lingjie.liu@seas.upenn.edu*
*University of Pennsylvania*

**Reviewed on OpenReview:** *https://openreview.net/forum?id=GVizav9Zf8*

## Abstract

Recent advances in 2D image generation have achieved remarkable quality, largely driven by the capacity of diffusion models and the availability of large-scale datasets. However, direct 3D generation is still constrained by the scarcity and lower fidelity of 3D datasets. In this paper, we introduce *Zero-1-to-G*, a novel approach that addresses this problem by enabling direct single-view generation on Gaussian splats using pretrained 2D diffusion models. Our key insight is that Gaussian splats, a 3D representation, can be decomposed into multi-view images encoding different attributes. This reframes the challenging task of direct 3D generation within a 2D diffusion framework, allowing us to leverage the rich priors of pretrained 2D diffusion models. To incorporate 3D awareness, we introduce cross-view and cross-attribute attention layers, which capture complex correlations and enforce 3D consistency across generated splats. This makes *Zero-1-to-G* the first direct **image-to-3D** generative model to effectively utilize **pretrained** 2D diffusion priors, enabling efficient training and improved generalization to unseen objects. Extensive experiments on both synthetic and in-the-wild datasets demonstrate superior performance in 3D object generation, offering a new approach to high-quality 3D generation. Project page: https://mengxuyigit.github.io/projects/zero-1-to-G/

## 1 Introduction

Single image to 3D generation is a pivotal challenge in computer vision and graphics, supporting various downstream applications such as virtual reality and gaming technologies. A primary difficulty lies in managing the uncertainty of unseen regions, as these areas represent a conditional distribution based on the visible portions of a 3D object. Recent advancements in diffusion models (Ho et al., 2020; Rombach et al., 2022) have demonstrated significant efficacy in capturing complex data distributions within images and videos,

---

\*Equal contribution.

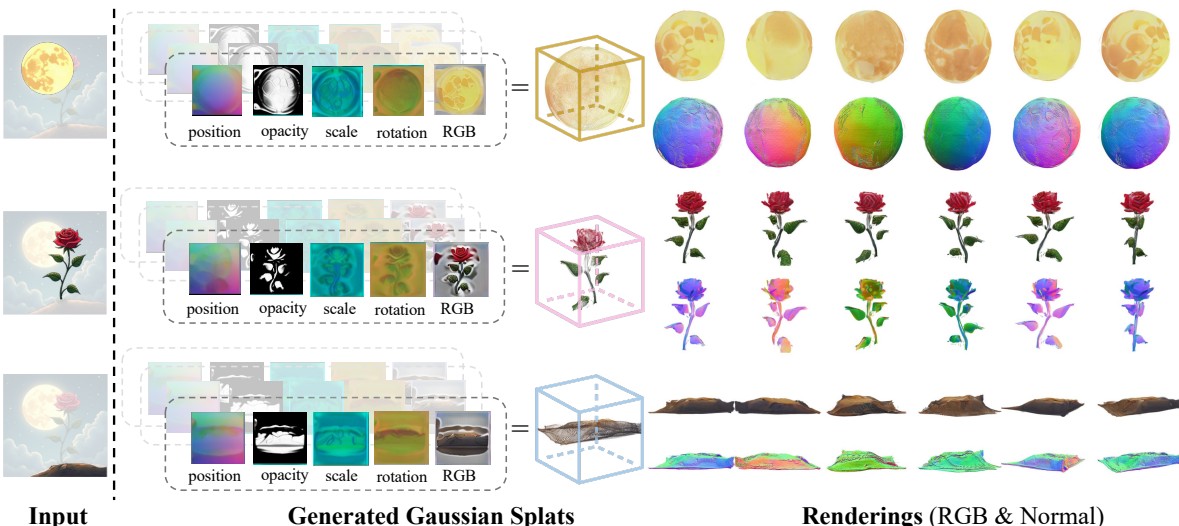

Figure 1: **Zero-1-to-G** tackles direct Gaussian splat generation from single images. By using pretrained 2D diffusion models, we are able to generalize to in-the-wild objects.

prompting researchers to harness these models for single image to 3D generation. Early efforts distilled 3D neural fields from pretrained 2D diffusion models via score distillation (Poole et al., 2022; Wang et al., 2023). However, these approaches necessitate per-scene optimization, which is time-consuming and susceptible to multi-faced Janus problems. Subsequent research achieved feed-forward generation by fine-tuning pretrained models to generate multi-view images of the same object (Liu et al., 2023b; Shi et al., 2023a; Long et al., 2023; Liu et al., 2023c) and enabling indirect 3D generation through sparse-view reconstruction models (Li et al., 2023; Tang et al., 2024; Xu et al., 2024), and Cat3D (Gao et al., 2024) further extends sparse view generation to dense view generation for better reconstruction. Although these two-stage methods enhance quality and efficiency, they often yield poor geometric fidelity and blurry renderings due to inconsistencies in multi-view images. To circumvent these limitations, recent methodologies have trained diffusion models directly on 3D representations (Liu et al., 2023d; Chen et al., 2023a; Zhang et al., 2024a; He et al., 2024; Nichol et al., 2022), thereby eliminating the reliance on multi-view images. However, direct 3D generation techniques necessitate training from scratch, requiring substantial computational resources and large 3D datasets, which remain scarce—three orders of magnitude less prevalent than 2D data.

In this paper, we propose a novel approach for direct 3D generation that unites the strengths of both worlds: it leverages the expressive power of 2D diffusion networks while maintaining the 3D structural consistency required for accurate 3D generation. Our key contribution is bridging the gap between Gaussian splats and natural images typically used in 2D generation tasks. While the original Gaussian splats consist of 14-channel images encoding various attributes, we decompose each of them into multiple 3-channel attribute images while preserving its 3D information (Sec. 3.1) This decomposition also enables efficient latent diffusion training by projecting the splatter images into the latent space of a pretrained VAE, making our method directly generate 3D structures within a pretrained 2D diffusion framework. To further align the latent space of pretrained VAE for efficient 3D information reconstruction, we fine-tune the VAE decoder to address the domain gap between splatter images and natural images, as we observed that splatter image quality is highly sensitive to pixel-level variations (Sec. 3.2). To ensure consistency among all the generated images representing different attributes under different views, we introduce multi-view attention layers and multi-attribute attention layers respectively into the Stable Diffusion for information exchange (Sec. 3.3).

It is important to note that, although we generate multiview splatter images, our method produces more structurally consistent and higher-quality results compared to traditional two-stage multiview 3D generation approaches. In two-stage methods (Xu et al., 2024; Tang et al., 2024), strong pixel-level consistency is required in the first stage to ensure accurate reconstruction in the second stage, which is often difficult to achieve. Moreover, the first stage operates independently of the second, lacking coordination between the two, thereby exacerbating inconsistencies. In contrast, our approach directly generates the final 3D representation

in a single stage, eliminating the need for pixel-level correspondence between multiview splatter images. Since splatter images can contain redundant information, and their spatial positions do not necessarily map to the final 3D positions, this single-stage process offers greater flexibility and robustness, ensuring a more consistent final 3D structure.

By leveraging pretrained 2D diffusion priors, our method not only improves training efficiency compared to existing 3D generation methods (Tab.2) but also exhibit strong diversity (Fig.12) and generalizes better to in-the-wild data (Fig.5) and real-world data (Fig.10). Overall, our contributions can be summarized as below:

- We present *Zero-1-to-G*, a novel direct 3D generative model for Gaussian splats that achieves excellent 3D consistency, diversity and superior rendering quality.
- We observe that Gaussian splats, as a 3D representation, can be decomposed into a set of 2D images of different views and attributes, making them inherently compatible with 2D image generation frameworks.
- Through decomposition and transformation of splatter images, we use 2D diffusion models for direct 3D generation with proper fine-tuning, unleash the power of pretrained 2D diffusion for training efficiently and better generalization towards in-the-wild data.

## 2 Related Works

To tame 2D diffusion for 3D generation, it is important to verify that **2D diffusion contains 3D priors**. Based on that, 3D diffusion is approached via two different ways: **distilling 3D representation by optimization**, or **generate multi-view images as an intermediate step**. While they benefit from the pretrained 2D diffusion prior, they are not as consistent as **direct 3D generation**, which in turn has its own limitations. Therefore we would like to combine the good of both worlds to achieve both efficiency and consistency in 3D generation. At the end, we have discussed some concurrent works that also use **2D diffusion for direct 3D generation**, which have similar pipelines but are different in high-level motivations and inplementations.

**Rich 3D Priors in Pretrained 2D Diffusion** Pretrained 2D diffusion models Podell et al. (2023) trained on Internet-scale datasets exhibit a certain level of 3D understanding. Liu et al. (2023b) demonstrated their capability to capture viewpoint changes and fine-tuned them on large 3D multi-view datasets for novel view synthesis. Other works extend their utility beyond natural images to generate 3D-aware outputs such as depth and normals Zhang et al. (2023b); Ke et al. (2023); Long et al. (2023); Fu et al. (2024). Following this line of works that adapt 2D diffusion priors for 3D-aware image prediction, our method leverages pretrained 2D diffusion models to generate Gaussian splats as splatter images, improving the generalization ability of direct 3D generation.

**Optimization-based 3D Generation via Distillation** Dreamfusion (Poole et al., 2022) and subsequent works (Lin et al., 2023; Chen et al., 2023b; Wang et al., 2023) utilize a pretrained text-to-image diffusion model to optimize a 3D representation through score distillation. DreamGaussian (Tang et al., 2023) significantly reduces training time by optimizing Gaussian splats. However, score distillation-based methods still require minutes of optimization per scene, as they must compare renderings with diffusion outputs from various viewpoints, which limits their generation speed. Additionally, these methods lack a clear understanding of geometry and viewpoint, resulting in multi-face problems.

**Two-stage 3D Generation via Multi-view Diffusion** Researchers have opted to train reconstruction-based models for highly efficient 3D generation (Hong et al., 2023; Tochilkin et al., 2024; Woo et al., 2024; Zou et al., 2023; Xu et al., 2023). LRM (Hong et al., 2023) and TripoSR (Tochilkin et al., 2024) introduced a transformer-based model that directly output a triplane from a single image. The model is trained on million-scale data by comparing the renderings of the triplane with ground truth using regression-based loss. TriplaneGaussian (Zou et al., 2023) further used a hybrid triplane-Gaussian representation to greatly accelerate the rendering of the generated 3D assets. However, the main drawback of regression-based methods is their failure to account for the uncertainty in single-view to 3D generation. GECO (Wang et al., 2024a) attempts to address this issue by distilling knowledge from multi-view diffusion models into a feedforward model. CRM (Wang et al., 2024b) introduces a geometric prior by employing two separate diffusion models to generate orthographic RGB images and Canonical Coordinate Maps, which are then fused into a triplane

representation using a convolutional UNet to produce the final textured mesh. Nonetheless, their results remain limited by the quality of the generated multi-view images.

**One-stage (Direct) 3D Generation via 3D Diffusion** Significant efforts have been made to directly train diffusion models on various 3D representations, including point clouds (Luo & Hu, 2021; Zhou et al., 2021; Nichol et al., 2022; Jun & Nichol, 2023), meshes (Liu et al., 2023d), and neural fields (Chen et al., 2023a; Shue et al., 2023; Müller et al., 2023; Gupta et al., 2023). However, these methods are typically constrained to category-level datasets and often struggle to generate high-quality assets. More recent approaches have begun encoding 3D assets into more compact latent representations (Zhang et al., 2023a; Lan et al., 2024; Zhao et al., 2023; Zhang et al., 2024b; Hong et al., 2024; Dong et al., 2024), enabling diffusion models to be trained more efficiently and enhancing generalization capabilities. Despite these advancements, direct 3D generative models are still primarily trained on synthetic 3D datasets like Objaverse (Deitke et al., 2024), which may hinder their ability to effectively handle more in-the-wild inputs.

More closely related to our work are GVGen (He et al., 2024) and GaussianCube (Zhang et al., 2024a), which also train diffusion models to generate Gaussian splats. Different from our method of leveraging 2D diffusion prior, they learn a latent space from scratch by directly encoding a structured volume of Gaussian splats, limiting their generalization ability.

**Direct 3D Generation via 2D Diffusion** Several works (Yan et al., 2024; Elizarov et al., 2024; Wu et al., 2024) also find the potential of tuning 2D diffusion to acquire 3D understanding and propose methods of generating 3D with 2D diffusion models. Specifically, Omage (Yan et al., 2024) uses a 12-channel UV atlas, requiring training from scratch without leveraging pretrained 2D diffusion priors, limiting its generalization ability. On the other hand, GIMDiffusion (Elizarov et al., 2024) decomposes the UV atlas into separate geometry maps and albedo textures to match the 3-channel output, but the use of pretraiend 2D diffusion models are limited to albedo generation. Still, both approaches focus on text-to-3D generation while we focus more on single-view image-to-3D reconstruction, and their reliance on mesh representations limits flexibility to model real-world data containing complex backgrounds, while our approach offers greater adaptability to diverse scenarios. While both our method and Wu et al. (2024) adopt 3D Gaussian Splatting as the final representation, our approaches differ significantly in both design and motivation. At the design level, Wu et al. (2024) uses a 2D diffusion model to predict latents for only RGB and depth, then extends the decoder to infer all Gaussian attributes from these two modalities. On the other hand, we directly predict latents for each individual Gaussian attribute and adopt a cross-domain mechanism to ensure their consistency, and keep the decoder architecture unchanged to largely preserve the pretrained latent space. These different design choices stem from different motivations. Our insight is that each individual Gaussian attribute image exhibits structural similarities to natural RGB images, which motivates our method to treat each attribute image as the generation target, enabling the 2D diffusion model to leverage its strong prior across all attribute domains. The decoder then processes these attribute images just like standard RGB images. On the other hand, Wu et al. (2024) relies more heavily on the decoder to interpret and reconstruct all Gaussian attributes, placing less emphasis on adapting the diffusion prior for understanding these domains.

# 3 Methods

Our method  Zero-1-to-G is a single stage direct 3D generation: given single view input **I**,  Zero-1-to-G generates the corresponding 3D representation **z**, where $z = \{z_i | i = 1, ..., N\}$ multiple Splatter Images under **N** camera views.

To harness the power of large-scale pretrained 2D diffusion models for direct 3D generation, we represent each 3D object as a set of multi-view Splatter Images (Szymanowicz et al., 2023). In Sec. 3.1, we detail our decomposition process, converting each multi-view splatter image into five 2D attribute images corresponding to RGB color, scale, rotation, opacity, and position. This decomposition allows us to effectively leverage the priors of 2D pretrained diffusion models to learn the underlying 3D object distribution (Sec. 3.3). Furthermore, we fine-tune the VAE decoder to enhance the rendering quality of the decoded Splatter Images (Sec. 3.2).

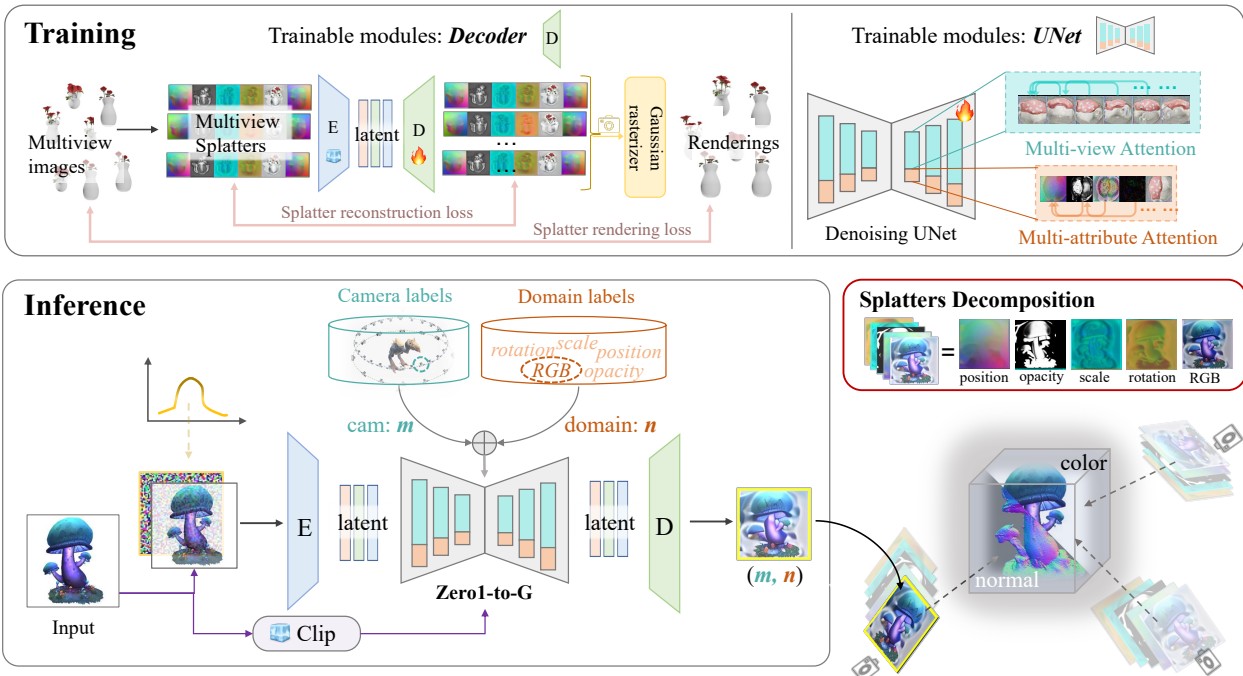

Figure 2: The pipeline of Zero-1-to-G. During training, we fine-tune both the VAE decoder (Sec. 3.2) and the denoising UNet (Sec. 3.3) of Stable Diffusion. At inference time, given a single view input of the target object, each component in the splatter image is generated by conditioning the camera view and the attribute switcher. The generated set of splatter image components can be directly fused into Gaussian splats (Sec. 3.1). Here we show splatter images of 3 views for better illustration, while our main experiments are conducted with 6 views.

## 3.1 Data Curation: Splatter Attribute Images

Gaussian splats can be rearranged into $H \times W$ grids, called Splatter Images (Szymanowicz et al., 2023), with 14 channels stacking five Gaussian attributes. Each attribute has a defined physical meaning and a degree of freedom of either 1 or 3, allowing representation as 3-channel RGB images. Thus, a Splatter Image can be decomposed into multiple attribute images, as detailed in Appendix A.1. An example of this transformation is shown in Fig.2, where each attribute image resembles a stylized RGB image. We further assessed the reconstruction fidelity on attribute images using a pretrained VAE, leading to our key observation that all attribute images are well modeled within the distribution of the pretrained 2D diffusion models, enabling their use for generating Splatter Images.

To train our generative model, we require high-quality ground-truth Splatter Images. Direct optimization per object introduces high-frequency artifacts due to the independent nature of Gaussian points, which pretrained VAEs struggle to reconstruct. Instead, we train a reconstruction network to generate smooth, well-regularized Splatter Images, avoiding artifacts while maintaining efficiency and scalability. This approach is significantly more effective than per-scene fitting, ensuring high-fidelity, diffusion-compatible representations for 3D generation.

## 3.2 Latent-space Alignment: VAE Decoder Fine-tuning

The pretrained VAE of Stable Diffusion is trained to reconstruct visually appealing images, though human eyes can hardly differentiate, and the 2D reconstruction metric is also good, there might be huge errors when rendering in 3D (Fig.7). While directly utilizing this VAE can reconstruct visually appealing Splatter Images, it does not guarantee high-quality RGB renderings and they often exhibit noticeable artifacts (Fig.8). These artifacts arise from two main factors: (1) each pixel in the splatter image corresponds to a Gaussian splat, meaning that even minor changes in pixel values can significantly affect the final rendering, and (2) Splatter Images contain high-frequency details that are challenging for the VAE to recover accurately. Therefore, we

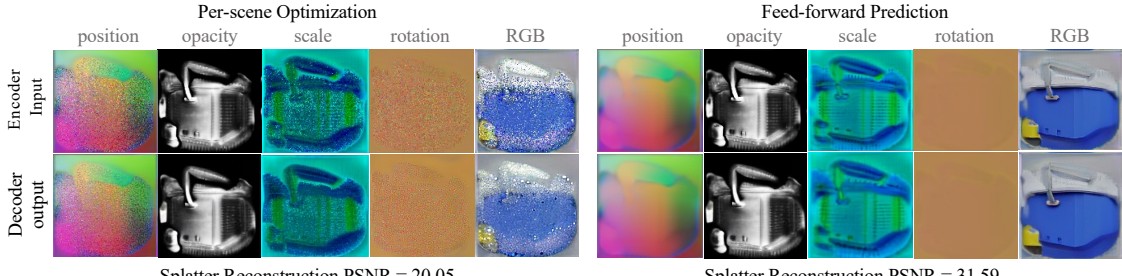

Figure 3: VAE encoding and decoding comparison with per-scene optimized splatters and feed-forward predicted splatters.

need to finetune the VAE to make it aware of the 3D information. The question is: shall we finetune the whole VAE or only the decoder? To better preserve the original latent space in order to use the pretrained 2D diffusion model, we want to make minimal changes to the original latent distribution. Since the reconstruction results are quite good in the view of 2D space, we decided to only finetune the decoder of VAE to increase the accuracy in recovering the 3D information. Therefore, we finetuned the VAE to suit our specific use case scenario, while preserving its strong encoding ability. This practice is also proven effective in image and video diffusion models. For example, Stable Diffusion fine-tunes its VAEs on human image datasets to improve facial reconstruction, and video diffusion models (Blattmann et al., 2023) fine-tune VAE decoders to enhance temporal consistency. Likewise, our fine-tuning enhances the VAE's understanding of 3D information in Splatter Images, ensuring high-fidelity reconstruction and rendering.

Specifically, the rendering loss used for VAE decoder finetuning comprises two components: MSE loss and LPIPS loss, defined as follows:

$$\mathcal{L}_{\mathrm{rgb}} = \mathcal{L}_{\mathrm{MSE}} + \mathcal{L}_{\mathrm{LPIPS}}$$

The overall objective function of decoder finetuning is defined as:

$$\mathcal{L}_{\mathrm{decoder}} = \mathcal{L}_{\mathrm{splatter}} + \mathcal{L}_{\mathrm{normal}} + \mathcal{L}_{\mathrm{rgb}} + \mathcal{L}_{\mathrm{mask}} \tag{1}$$

where $\mathcal{L}_{\mathrm{splatter}}$ denotes the reconstruction loss of splatter image itself, while $\mathcal{L}_{\mathrm{normal}}$, $\mathcal{L}_{\mathrm{rgb}}$ and $\mathcal{L}_{\mathrm{mask}}$ are all about the renderings of the reconstructed splatter, denoting cosine similarity loss of rendered normals, the sum of all losses of rendered images, and the binary cross-entropy loss of the rendered masks.

### 3.3 Retargeting 2D Diffusion Output

With the decomposition discussed in Sec 3.1, we are ready to learn a 3D generative model with 2D diffusion framework by generating Gaussian splats represented by a set of multi-view attribute images. Specifically, the generation target of each object is sampled from a distribution $p(\mathbf{z})$, which is modeled as a joint distribution of $K$ splatter images under fixed camera views and $N$ splatter attributes under each camera view. With the fixed camera viewpoints $\{\boldsymbol{\pi}_1, \boldsymbol{\pi}_2, \cdots, \boldsymbol{\pi}_K\}$ and condition input image $y$, the object is modeled as:

$$p(\mathbf{z}) = p(\mathbf{z}^{(1:K,1:N)}|y) = p_{\mathrm{pos, op, sc, rot, rgb}} \left( \{\mathbf{z}_{\mathrm{pos}}^{1:K}, \mathbf{z}_{\mathrm{op}}^{1:K}, \mathbf{z}_{\mathrm{sc}}^{1:K}, \mathbf{z}_{\mathrm{rot}}^{1:K}, \mathbf{z}_{\mathrm{rgb}}^{1:K} \} \mid y \right)$$

where $\{pos, op, sc, rot, rgb\}$ are the $N = 5$ attributes of the splatter image.

To control which attribute and which camera view of the splatter images to generate, we need to incoporate both attribute condition and camera condition in addition, and we achieve this by concatenating 1-D embedding to the original time embedding during the diffusion process.

To further ensure the consistency among the generated attribute images of all camera views and all attributes, we insert additional attention layers into the pretrained diffusion UNet blocks to model the joint distribution of both camera views and Gaussian attributes of our decomposed Splatter Images. Please refer to Appendix A.1.3 for detailed implementation of multi-view attention and multi-attribute attention.

**Training Loss** During training, we organize each view and attribute of a splatter image within the batch dimension and apply independently sampled Gaussian noise. In each attention block, we alternately apply multi-view attention and multi-attribute attention to enhance the model's ability to learn complex correlations.

The forward process of our diffusion model is directly extended from the original DDPM (Ho et al., 2020), which is

$$q(\mathbf{z}_{1:T}^{(1:K,1:N)}|\mathbf{z}_0^{(1:K,1:N)}) = \prod_{t=1}^{T} q(\mathbf{z}_t^{(1:K,1:N)}|\mathbf{z}_{t-1}^{(1:K,1:N)}) = \prod_{t=1}^{T} \prod_{k=1}^{K} \prod_{n=1}^{N} q(\mathbf{z}_t^{(k,n)}|\mathbf{z}_{t-1}^{(k,n)}),$$

And the reverse process will be

$$p_\theta(\mathbf{z}_{0:T}^{(1:K,1:N)}) = p(\mathbf{z}_T^{(1:K,1:N)}) \prod_{t=1}^{T} p_\theta(\mathbf{z}_{t-1}^{(1:K,1:N)}|\mathbf{z}_t^{(1:K,1:N)})$$

$$= p(\mathbf{z}_T^{(1:K,1:N)}) \prod_{t=1}^{T} \prod_{k=1}^{K} \prod_{n=1}^{N} p_\theta(\mathbf{z}_{t-1}^{(k,n)}|\mathbf{z}_t^{(1:K,1:N)})$$

where

$$p_\theta(\mathbf{z}_{t-1}^{(k,n)}|\mathbf{z}_t^{(1:K,1:N)}) = \mathcal{N}\big(\mathbf{z}_{t-1}^{(k,n)}; \boldsymbol{\mu}_\theta^{(k,n)}(\mathbf{z}_t^{(1:K,1:N)}, t), \sigma_t^2 \mathbf{I}\big)$$

The definition of the Gaussian mean for the reverse process is defined as:

$$\mu_\theta^{(k,n)}(\mathbf{z}_t^{(1:K,1:N)}, t) = \frac{1}{\sqrt{\alpha_t}} \left( \mathbf{z}_t^{(k,n)} - \frac{\beta_t}{\sqrt{1-\bar{\alpha}_t}} \epsilon_\theta^{(k,n)}(\mathbf{z}_t^{(1:K,1:N)}, t) \right)$$

The corresponding loss function for multi-view and multi-domain modeling is as follows:

$$\ell = \mathbb{E}_{t, \mathbf{x}_0^{(1:K,1:N)}, k, n, \epsilon^{(1:K,1:N)}} \left[ \|\epsilon^{(k,n)} - \epsilon_\theta^{(k,n)}(\mathbf{z}_t^{(1:K,1:N)}, t)\|_2^2 \right]$$

where $\epsilon^{(k,n)}$ is the Gaussian noise added to attribute $n$ for the $k$-th view, and $\epsilon_\theta^{(k,n)}$ is the model's noise prediction for attribute $n$ in the $k$-th view.

## 4 Experiments

### 4.1 Implementation Details

**Dataset** We train on the G-buffer Objaverse (Qiu et al., 2024) dataset, which consists of approximately 262,000 objects sourced from Objaverse (Deitke et al., 2024). Each object in the dataset is rendered from 38 viewpoints, with additional normal and depth renderings provided. For generating the ground truth splatter images, we use the first viewpoint as the input condition, along with five additional views at the same elevation and azimuth angles of 30°, 90°, 180°, 270°, and 330° to comprehensively cover the full 360 degrees. We only use the RGB and normal renderings for the supervision of decoder finetuning and Gaussian Splats prediction model.

**Model Training and Inference** We initialize our model from Stable Diffusion Image Variations. Following Wonder3D (Long et al., 2023), our training includes two stages. In the first stage, we only train multi-view attention, and in the second stage, we add one more cross-domain attention layer for training, and together fine-tune the multi-view attention layer learned in the first stage. For the first stage, we use a batch size of 64 on 4 NVIDIA L40 GPUs for 13k iterations, which takes about 1 day. For the second stage, we use a batch size of 64 on 8 NVIDIA L40 GPUs for 30k iterations, which takes about 2 days. For decoder fine-tuning, we use a total batch size of 64 on 8 NVIDIA L40 GPUs for 20k iterations. The second stage of training takes about 2 days. During inference, we use cfg = 3.5 and our method can generate Gaussian splats per object in 8.7 seconds on a single NVIDIA L40 GPU.

## 4.2 Evaluation Protocol

**Dataset and Metrics** Following prior works (Liu et al., 2023c;a; Wang et al., 2024a), we conduct quantitative comparisons using the Google Scanned Objects (GSO) dataset (Downs et al., 2022). Specifically, we utilize a randomly selected subset of 30 objects from the GSO dataset, including a variety of everyday items and animals, as in SyncDreamer (Liu et al., 2023c). For each object, a conditioning image is rendered at a resolution of $512 \times 512$ with an elevation angle of $10°$. Evaluation images are then generated at evenly spaced $30°$ azimuthal intervals around the object, keeping the elevation constant.

To assess the quality of novel view synthesis, we report standard metrics such as PSNR, SSIM (Wang et al., 2004), and LPIPS (Zhang et al., 2018). Additionally, we evaluate the geometry of our generated outputs using Chamfer Distance (CD). Please refer to Table 1 for details.

Beyond the GSO dataset, we also evaluate our approach on in-the-wild images to demonstrate its robustness and generalizability (Figure 5).

**Baselines** We compare our methods against several recent approaches across different categories. For reconstruction-based methods, we include TriplaneGS (Zou et al., 2023) and TripoSR (Tochilkin et al., 2024). In the realm of direct 3D generation, we compare with LN3Diff (Lan et al., 2024). Finally, for two-stage methods transitioning from single-image to multi-view to 3D, we include InstantMesh (Xu et al., 2024) and LGM (Tang et al., 2024).

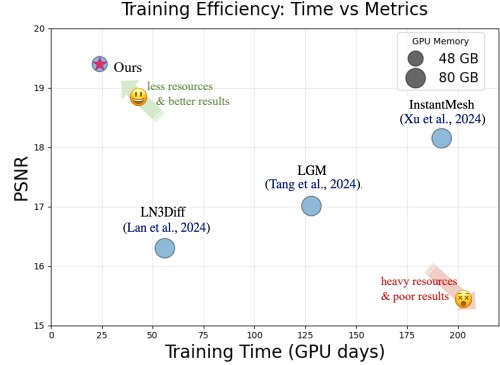

Figure 4: Model performance v.s. training resources: highlighting our method's superior training efficiency and accuracy.

## 4.3 Results

**Qualitative Comparison** Figure 5 showcases rendering results of Zero-1-to-G compared to several baselines on in-the-wild inputs. Two-stage methods such as LGM (Tang et al., 2024) and InstantMesh (Xu et al., 2024) often suffer from quality and consistency issues due to their reliance on multi-view image generation. LGM frequently produces overly smooth textures with artifacts (e.g., blue backgrounds in the first and third examples) and inconsistent 3D Gaussians with "floaters" (fourth and fifth examples). InstantMesh generates more consistent outputs but exhibits noticeable smoothness and grid-like texture artifacts due to its reconstruction process. Both methods are ultimately constrained by the limitations of their multi-view generation stage, leading to flawed geometry and inconsistencies (e.g., the last example). In contrast, our method directly operates in 3D space and leverages pretrained 2D diffusion priors, enabling accurate geometry and consistent renderings. Another direct image-to-3D method, LN3Diff (Lan et al., 2024), struggles to capture fine-grained textures and tends to produce oversmoothed geometry. This is likely due to its training-from-scratch strategy, which limits generalization and performance on unseen real-world objects. Our approach, benefiting from one-stage direct generation and rich 2D priors, achieves higher fidelity in both geometry and texture, particularly on in-the-wild inputs (Figure 5) and real-world data with background clutter (Figure 10).

**Quantitative Comparison** The quantitative results on the GSO dataset, presented in Table 1, show that Zero-1-to-G consistently outperforms all baselines across all metrics. Reconstruction-based methods, like TriplaneGaussian and TripoSR, struggle with sharp predictions for unseen regions due to their deterministic nature. Two-stage methods, such as InstantMesh, perform reasonably well but are still limited by sparse multi-view images. Direct 3D methods like LN3Diff underperform due to the lack of pretrained priors.

**Training Efficiency** By leveraging pretrained diffusion priors, our method reduces training time and resource requirements. We complete training in just 3 days using only 8 NVIDIA L40 GPUs, which is more efficient compared to previous two-stage and direct 3D generation methods, as detailed in Table 2. This efficiency highlights the advantage of integrating 2D priors for direct 3D generation, reducing the need for extensive computational resources.

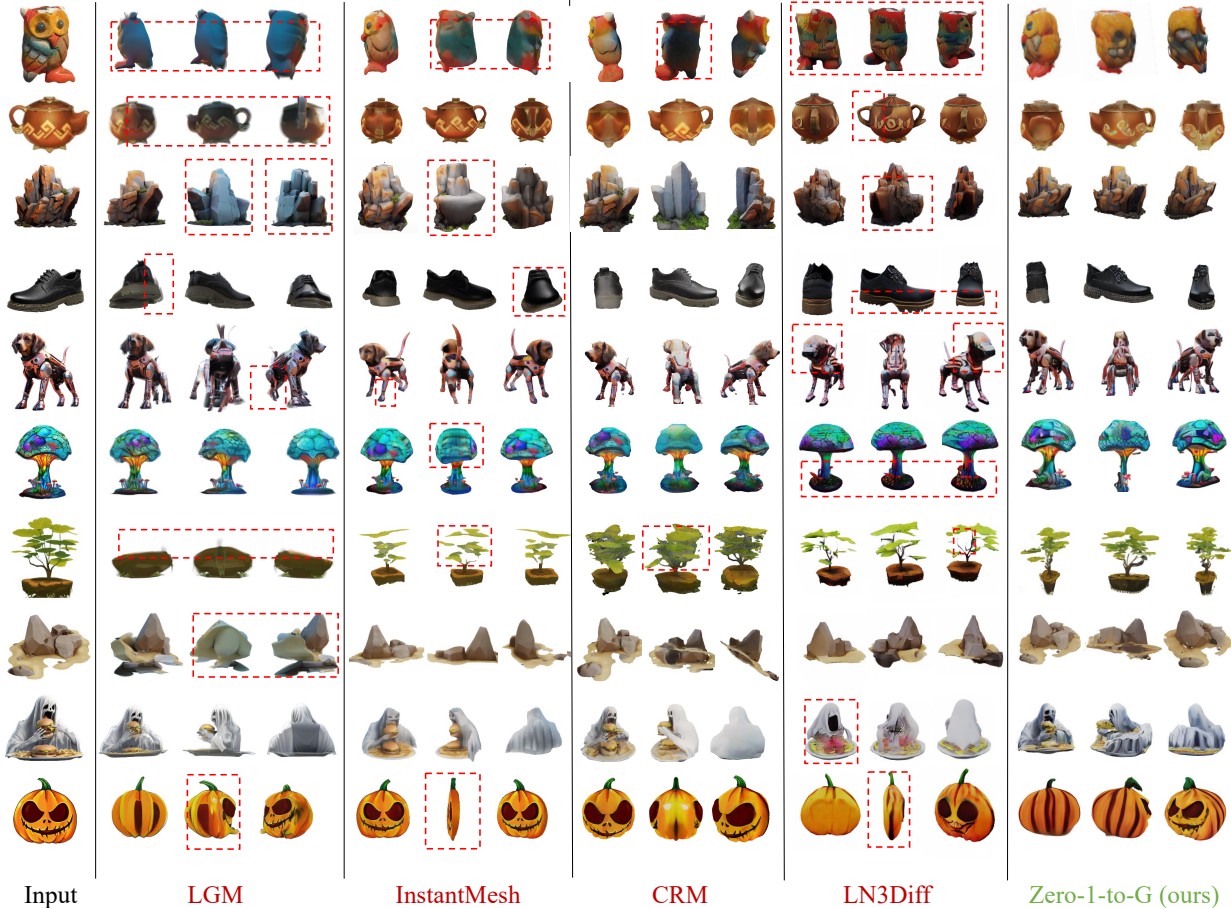

| | | | | | |
|---|---|---|---|---|---|
| Input | LGM | InstantMesh | CRM | LN3Diff | Zero-1-to-G (ours) |

Figure 5: Qualitative comparison with, LGM, InstantMesh, LN3Diff on in-the-wild input images.

Table 1: Quantitative comparison between our methods and other baselines on the GSO dataset.

| Methods | PSNR ↑ | SSIM ↑ | LPIPS ↓ | CD ↓ |
|---|---|---|---|---|
| TriplaneGS | 17.80 | 0.811 | 0.216 | 0.0440 |
| TripoSR | 17.32 | 0.804 | 0.217 | 0.0423 |
| LGM | 17.01 | 0.793 | 0.199 | 0.0621 |
| InstantMesh | 18.15 | 0.810 | 0.179 | 0.0419 |
| LN3Diff | 16.30 | 0.786 | 0.241 | 0.0637 |
| CRM | 16.10 | 0.818 | 0.208 | 0.0423 |
| Ours (10 steps) | 19.03 | 0.812 | 0.182 | 0.0396 |
| Ours (35 steps) | **19.40** | **0.818** | **0.178** | **0.0390** |

## 4.4 Ablation Study

**VAE Decoder Finetuning** Without fine-tuning VAE decoder, although the decoded splatter image visually looks identical to the original input (Fig.9), the renderings exhibit noticeable artifacts (Fig.6). Since each pixel represents a Gaussian splat and the decoder cannot capture high-frequency areas, well-reconstructed splatter images don't necessarily ensure good renderings. Fig.6 *full model* shows the effectiveness of preserving high-frequency 3D information in splatter images by finetuning the decoder.

**Cross-attribute Attention** We can see from Figure 6, if we don't use cross-attribute attention, the renderings of the Gaussian splats have many floaters and the textures become blurry, this is because that different attributes of the same Gaussian splat are not well aligned.

Table 2: Comparison of training efficiency with other baseline methods. For LGM and InstantMesh, we only count the reconstruction part (indicated by †), since their multi-view diffusion module directly take pretrained models, otherwise more time and resources are needed to the multi-view generation part.

| Methods | Training Time ↓ | GPUs ↓ |
|---|---|---|
| LGM † | 4 days | 32 * A100 (80G) |
| InstantMesh † | 12 days | 16 * H800 (80G) |
| LN3Diff | 7 days | 8 * A100 (80G) |
| Ours | 3 days | 8 * L40 (48G) |

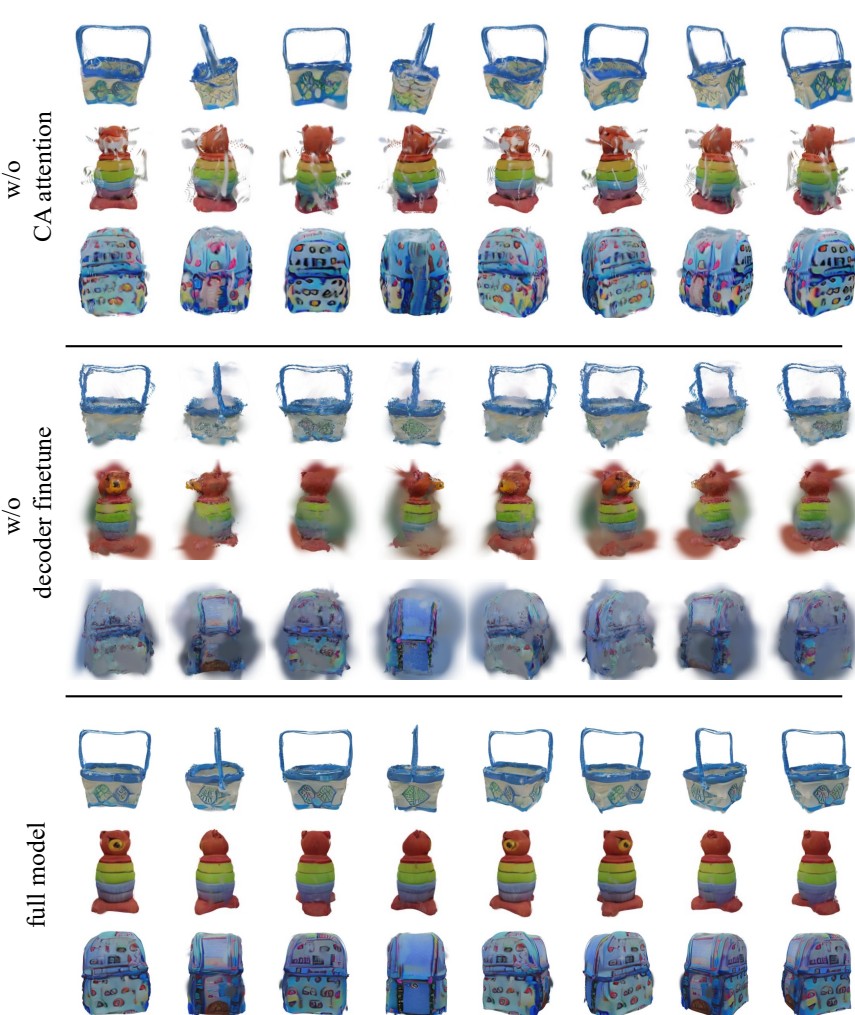

Figure 6: Ablation study on GSO dataset.

**No Diffusion Prior** The use of diffusion prior is essential in our model. To verify this, we conducted training with the same StableDiffusion UNet architecture but using random initialization, and we also added the same cross-view attention layers to the UNet as in our method. We can see that without using the prior, the training of the model cannot converge to meaningful results using the same data and training iterations.

**Number of Views and Resolutions of the Generated Splatters** Since our method generates 3D Gaussians through multi-view splatter images, the quality of the resulting representation can be influenced by two main factors: 1) the total number of views in the multi-view splatters, and 2)the resolution of each view. While increasing the number of Gaussian splats can enhance the expressiveness of the 3D Gaussian

Table 3: Ablation study on module design, inference with GSO dataset.

| Model Components | PSNR ↑ | SSIM ↑ | LPIPS ↓ |
|---|---|---|---|
| w/o diffusion prior | 9.39 | 0.592 | 0.722 |
| w/o decoder fine-tuning | 17.13 | 0.775 | 0.272 |
| w/o cross-attribute attention | 17.26 | 0.767 | 0.237 |
| Full Model | **19.40** | **0.818** | **0.178** |

representation, it may also introduce redundancy and cross-view inconsistency in the absence of a proper pruning strategy during the feedforward generation. Such issues can lead to degradation in rendering quality.

To ensure fair and controlled comparisons across all ablation settings, we reduced the training configuration by limiting both the dataset size and the training iterations. It is important to note that the results in this ablation study are intended solely to analyze the effect of view count and resolution on splatter quality within our method. These models are not fully trained and should not be directly compared against other baselines.

Specifically, we used a subset of 26,000 objects from the full dataset and trained for 20,000 iterations for both Stage 1 (multi-view training) and Stage 2 (multi-domain training), using batch sizes of 128 and 64, respectively.

For evaluation, we used classifier-free guidance with CFG=3.0 and 30 denoising steps on the GSO dataset. The results are reported in Table 4.

We observed that higher-resolution splatters ($256 \times 256$) indeed lead to sharper and more detailed textures in the rendered Gaussian splats. However, they also occasionally introduce more floaters around the objects, particularly for objects with thin structures. This artifact appears to stem from non-foreground Gaussians with relatively high opacity. Although the higher-resolution setting tends to yield more visual artifacts within the same training budget and results in a negative impact on quantitative metrics, we found that extending the training time helps mitigate this issue to some extent. This suggests that with sufficient training, using higher-resolution splatter images has the potential to significantly improve rendering quality.

Looking ahead, it is worth exploring strategies for both training and inference with higher-resolution splatters. A more stable and efficient training scheme could involve progressive resolution scaling: starting with low-resolution splatters to capture geometry and coarse textures, then gradually increasing the resolution to refine fine-grained details and sharp structures. On the inference side, the main challenge lies in efficiency, as higher resolutions substantially increase runtime. To address this, one promising direction is to adopt diffusion distillation strategies such as in (Wang et al., 2024a), which can distill model knowledge and enable inference in one diffusion step. Together, these strategies highlight promising future directions for advancing high-resolution splatter image generation.

Table 4: Quantitative comparison on the number of views and resolutions in the generated splatters.

| Splatters Size (V×H×W) | PSNR ↑ | SSIM ↑ | LPIPS ↓ | Inference Speed(s) ↓ |
|---|---|---|---|---|
| 4×128×128 | 18.02 | 0.832 | 0.191 | 6.59 |
| 6×128×128 | 18.21 | 0.832 | 0.189 | 8.46 |
| 4×256×256 | 13.92 | 0.726 | 0.434 | 17.92 |

## 5 Conclusion

In this work, we introduce a novel framework that leverages 2D diffusion priors for direct 3D generation by decomposing Gaussian splats into multi-view attribute images. This decomposition preserves the full 3D structure while efficiently mapping it to 2D images, enabling fine-tuning of pretrained Stable Diffusion models with cross-view and cross-attribute attention layers. Our approach significantly reduces computational costs compared to other direct 3D generation methods. By bypassing the stringent requirement for multi-view image consistency in two-stage approaches, we generate more accurate 3D geometry and produce higher-quality renderings through a single-stage diffusion process. Furthermore, our method exhibits stronger generalization

capabilities than existing direct 3D generation techniques due to the use of diffusion priors, offering a more efficient and scalable solution for 3D content creation.

**Limitations & Future Works** Despite achieving superior reconstruction metrics and strong generalization to in-the-wild data, our method has some limitations. First, our inference speed is not as fast as regression models, as each splatter must be generated through diffusion. A potential improvement would be to integrate a diffusion distillation (Song et al., 2023; Wang et al., 2024a; Gu et al., 2024) to reduce denoising steps. Second, we do not currently disentangle material and lighting conditions, leading to highlights and reflections being baked into the Gaussian splat texture. Future work could address this by incorporating inverse rendering to better predict non-Lambertian surfaces.

## Broader Impact Statement

This work advances conditional generation of 3D assets from single images, which may benefit applications such as game design, movie production, and virtual reality by reducing manual modeling effort. However, the ability to generate 3D objects from real images also raises concerns about potential misuse in replicating humans or copyrighted content without consent. Our study is conducted on publicly available object datasets, and we encourage responsible use of this technology in practice.

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

# A   Appendix

## A.1   Implementation Details

### A.1.1   Splatter Image Transformation

Each attribute, except opacity, possesses three degree-of-freedoms, which align gracefully with the 3 channels of the RGB space. The following illustrates the detailed operations to convert each attribute into an RGB image

- RGB: the RGB attribute naturally lies in the RGB space, no conversion is needed.

- Position: We normalize the 3D object in the bounding box $[-1, 1]$ and use the 3D coordinates of each Gaussian as position attribute.

- Scale: The raw scale value spans from $1e-15$ to $1e-2$, so directly converting the 3D scale to RGB space using the min-max value for the whole dataset will result in most regions being zeros due to the significant difference in power. The value distribution also does not match the normal distribution, making it difficult for diffusion models to learn effectively. We thus convert the raw scale values to log-space and clamp the minimum values to $-10$, as we found Gaussian splats with scales smaller $1e-10$ will have negligible effects on the final rendering.

- Rotation: We first convert the 4-dimensional quaternion to 3-dimensional axis angle, then normalize it to $[-1, 1]$.

- Opacity: We directly duplicate the single channel to 3 channels, and average the predicted 3-channel image to get the final opacity prediction.

To obtain the final 3D representation from the decomposed attribute images, we first revert each attribute to its original scale and number of channels, then concatenate all attributes channel-wise to form a single splatter image per view. To fuse the multi-view splatter images, we take the union of all Gaussians across views. While this may introduce overlapping Gaussians corresponding to the same 3D region, our method does not require explicit pixel-level correspondence across views.

### A.1.2   VAE Decoder Finetuning

We use two toy examples—a 2D natural image and a Splatter image—to demonstrate the necessity of fine-tuning the VAE decoder for preserving high-quality rendering of Splatter images.

When reconstructing 2D natural images with a VAE, the visual quality is generally acceptable, but the pixel-space error is significant, as shown in Fig.7. While this error is tolerable for 2D reconstruction, it leads to noticeable artifacts when propagated to 3D. Fig.8 illustrates the issue with reconstructed Splatter images using the same pretrained VAE, where the reconstruction of the position attribute (*pos*) is notably inaccurate due to the VAE not being trained for 3D-aware images with high-frequency details.

Therefore, fine-tuning the VAE decoder is essential for accurate reconstruction of 3D-aware images. To preserve the pretrained latent space distribution and minimize changes to maintain compatibility with the 2D diffusion model, we freeze the VAE encoder and fine-tune only the decoder with an additional rendering loss, as described in Eq.1.

### A.1.3   UNet Fine-tuning

We fine-tune the UNet by inserting additional attention layers: **multi-view attention** and **multi-attribute attention**. When fine-tuning the StableDiffusion UNet, for both stages, we use a constant learning rate of $1e-4$ with a warmup of the first 100 steps. We use the Adam optimizer for both stages and the betas are set to $(0.9, 0.999)$. For classifier-free guidance, we drop the condition image with a probability of 0.1.

**Modeling Multi-View Distribution** Prior works have approached multi-view diffusion either by reshaping the batch dimension into a token dimension and applying self-attention (Shi et al., 2023b; Liu et al., 2023c; Long et al., 2023; Liu et al., 2024), or by spatially concatenating multi-view images to form a larger image, which directly maps the latent distribution to a multi-view distribution (Shi et al., 2023a). We choose the former approach for its flexibility in reshaping data for both cross-view and cross-attribute attention mechanisms. This design allows for efficient information exchange among different views, where tokens corresponding to the same attribute from different views are concatenated for self-attention. This facilitates our model's ability to learn a consistent multi-view distribution for each Gaussian attribute.

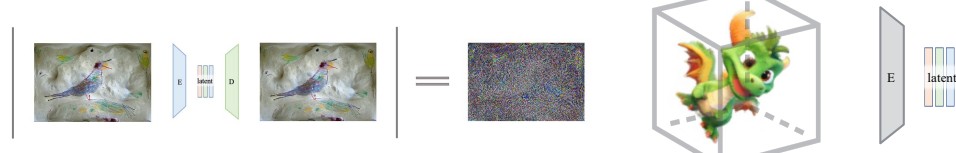

Figure 7: Error map of 2D image reconstruction using pretrained VAE of Stable Diffusion.

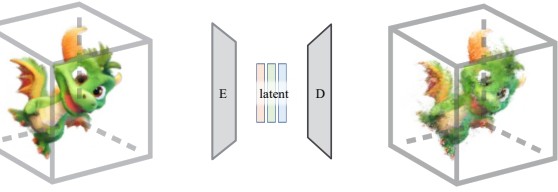

Figure 8: Rendering results of Splatter image reconstruction using pretrained VAE of Stable Diffusion.

**Modeling Multi-Attribute Distribution** Similar to (Long et al., 2023), we utilize an attribute switcher to specify which attribute the network should generate. To maintain consistency across generated images that represent different attributes of the same object, we employ an attention mechanism to capture the interactions between images taken from the same viewpoint but corresponding to different attributes.

Specifically, we introduce additional self-attention modules to model the cross-attribute correlations, where tokens representing all attributes from the same viewpoint are combined and processed using standard scaled dot-product attention.

More visualizations on ablation in Figure 9 shows the splatter image visualization of ablation study. We can see that without cross-attribute attention, there are obvious misalignments of different domains of the splatter images. Without decoder fine-tuning, although the splatter image is visually good, the rendering is not satisfying because Gaussian splats are sensitive to the value changes in the pixels. Fine-tuning the decoder can greatly improve the rendering quality.

**Statistics of Learned Latent Distribution** To evaluate the effectiveness of our method in accurately learning the underlying latent distribution of splatter images through fine-tuning from a pretrained 2D diffusion model, we computed the sample mean ($\mu$) and standard deviation ($\sigma$) of the ground truth splatters and compared them to those of the generated samples on the validation set after stage 2 training (cross-attribute attention). The comparison was conducted at a splatter resolution of $6 \times 128 \times 128$.

Table 5 presents the comparison of the latent statistics across different attributes. These results demonstrate that, even without explicit normalization of the latent space, our model is able to learn distinct and meaningful latent distributions for each attribute domain. This suggests that the model successfully aligns latent statistics in a data-driven manner during training.

To further quantify the similarity between the latent distributions of the ground truth and generated splatters, we computed the Bhattacharyya distance between the corresponding Gaussian distributions (parameterized by their means and standard deviations). The results are presented in Table 6.

These results validate that the learned latent distributions closely align with the ground truth, further confirming the model's effective convergence. While our approach does not require explicit latent normalization, this remains a promising avenue for future improvement, particularly for stabilizing training or adapting to novel domains.

### A.1.4 Splatter Image Reconstruction Network

To obtain the splatter image ground truth for our training, as mentioned in Sec. 3.1, we fine-tune LGM (Tang et al., 2024) to take as input 6 multi-view renderings of the G-Objaverse dataset and output splatter images

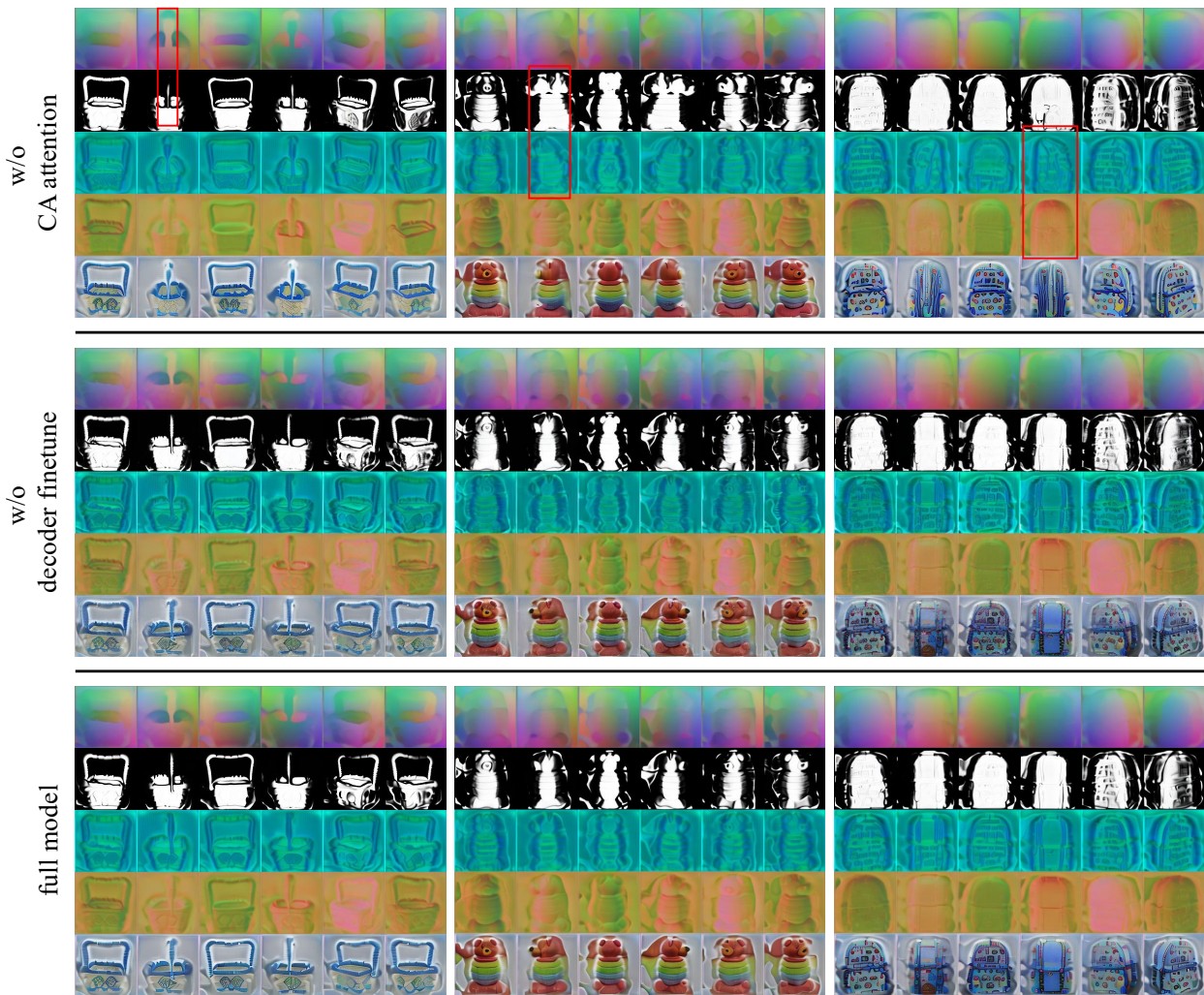

Figure 9: Splatter visualization of ablation study.

Table 5: Comparison of latent statistics (mean and standard deviation) between inference results and ground truth splatters for each attribute.

| Attribute | Inference | Ground Truth |
|---|---|---|
| pos | $\mu$: [0.5278, 0.5149, 0.5039]
$\sigma$: [0.1476, 0.1610, 0.1377] | $\mu$: [0.5238, 0.5055, 0.5012]
$\sigma$: [0.1495, 0.1639, 0.1398] |
| opacity | $\mu$: [0.1996, 0.1991, 0.1997]
$\sigma$: [0.3685, 0.3701, 0.3682] | $\mu$: [0.2056, 0.2056, 0.2056]
$\sigma$: [0.3741, 0.3741, 0.3741] |
| scale | $\mu$: [0.0003, 0.5513, 0.5742]
$\sigma$: [0.0036, 0.1066, 0.0994] | $\mu$: [0.0000, 0.5359, 0.5589]
$\sigma$: [0.0000, 0.1070, 0.1017] |
| rotation | $\mu$: [0.7119, 0.5185, 0.3040]
$\sigma$: [0.0637, 0.0332, 0.0785] | $\mu$: [0.7094, 0.5191, 0.2995]
$\sigma$: [0.0670, 0.0353, 0.0808] |
| rgbs | $\mu$: [0.5956, 0.5792, 0.5764]
$\sigma$: [0.1760, 0.1777, 0.1875] | $\mu$: [0.5832, 0.5678, 0.5699]
$\sigma$: [0.1746, 0.1754, 0.1837] |

of 2D Gaussian splatting (Huang et al., 2024). The training objective is to compare the splatter renderings with ground truth images using MSE and LPIPS loss. We also use cosine similarity loss between ground truth normals and rendered normals. We fine-tune LGM for 30k iterations with a batch size of 32 on 8 NVIDIA L40 GPUS, which takes about 1 day.

Table 6: Bhattacharyya distance between the ground truth and inference Gaussian distributions for each attribute. *Note: The first channel of the *scale* attribute is excluded from rendering and analysis.

| Attribute | Bhattacharyya Distance↓ |
|-----------|------------------------|
| pos | 0.000246 |
| opacity | 0.000084 |
| scale | 0.001860 |
| rotation | 0.000798 |
| rgbs | 0.000491 |

### A.2 More Results

**Results on Real-world Datasets** Benefit from the flexibility of Gaussian Splats, our method could be further extended to generate real-world scenes with backgrounds, which cannot be achieved by our concurrent methods using mesh-based representations. Below shows our testing results on real-world dataset MVImagenet Yu et al. (2023), which included diverse categories of different objects.

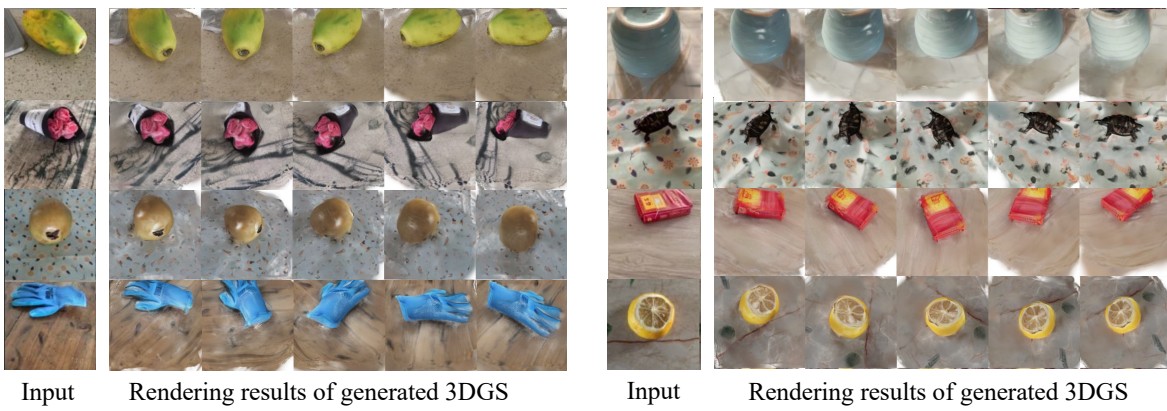

Input   Rendering results of generated 3DGS    Input   Rendering results of generated 3DGS

Figure 10: RGB and normal renderings of more examples on MVImgNet dataset.

**More examples** More RGB and normal renderings can be found in Figure 11 and Figure 10.

**Diversity** Since we model the unseen viewpoints with diffusion models, our results can generate diverse results given the same input Figure 12.

### A.3 Limitations and Future works

While our method demonstrates strong generalization and consistency across a wide range of examples, we also observe certain limitations that point to promising directions for future improvement. One notable failure case occurs in the generation of fine facial details, particularly when synthesizing novel views of human characters. For instance, in Figure 12, the generated front view of the girl character wearing blue clothing appears less consistent with the input view, especially in facial features.

We attribute this issue to limitations in the VAE used to encode the input image. Our method leverages a pretrained VAE from an earlier version of Stable Diffusion to extract latent features as conditioning input. However, these earlier VAEs are known to inadequately capture fine-grained facial details, which can lead to degraded performance in view synthesis tasks requiring high facial fidelity. Recent works have addressed this shortcoming by fine-tuning the Stable Diffusion VAE with additional facial-focused datasets and perceptual reconstruction losses, resulting in significantly better detail preservation for human faces.

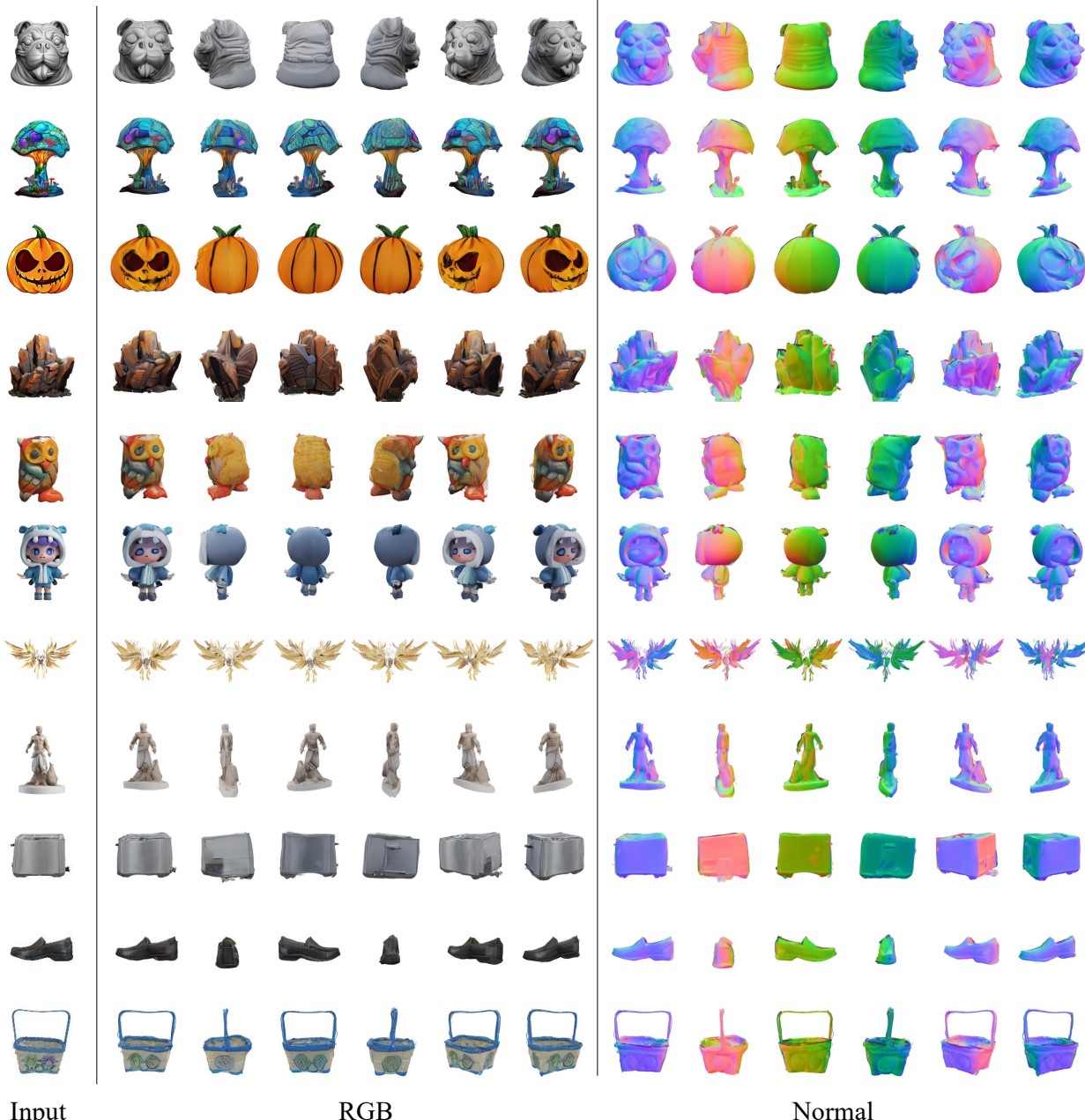

Input                               RGB                               Normal

Figure 11: RGB and normal renderings of more examples on in-the-wild and GSO datasets.

While adopting such enhanced VAE models could mitigate these failures, doing so may also reduce the model's ability to generalize across diverse object categories and styles. In this work, we intentionally chose to initialize from a baseline Stable Diffusion model to preserve broad generalization, which we believe is essential for our multiview synthesis framework to be effective across heterogeneous object domains.

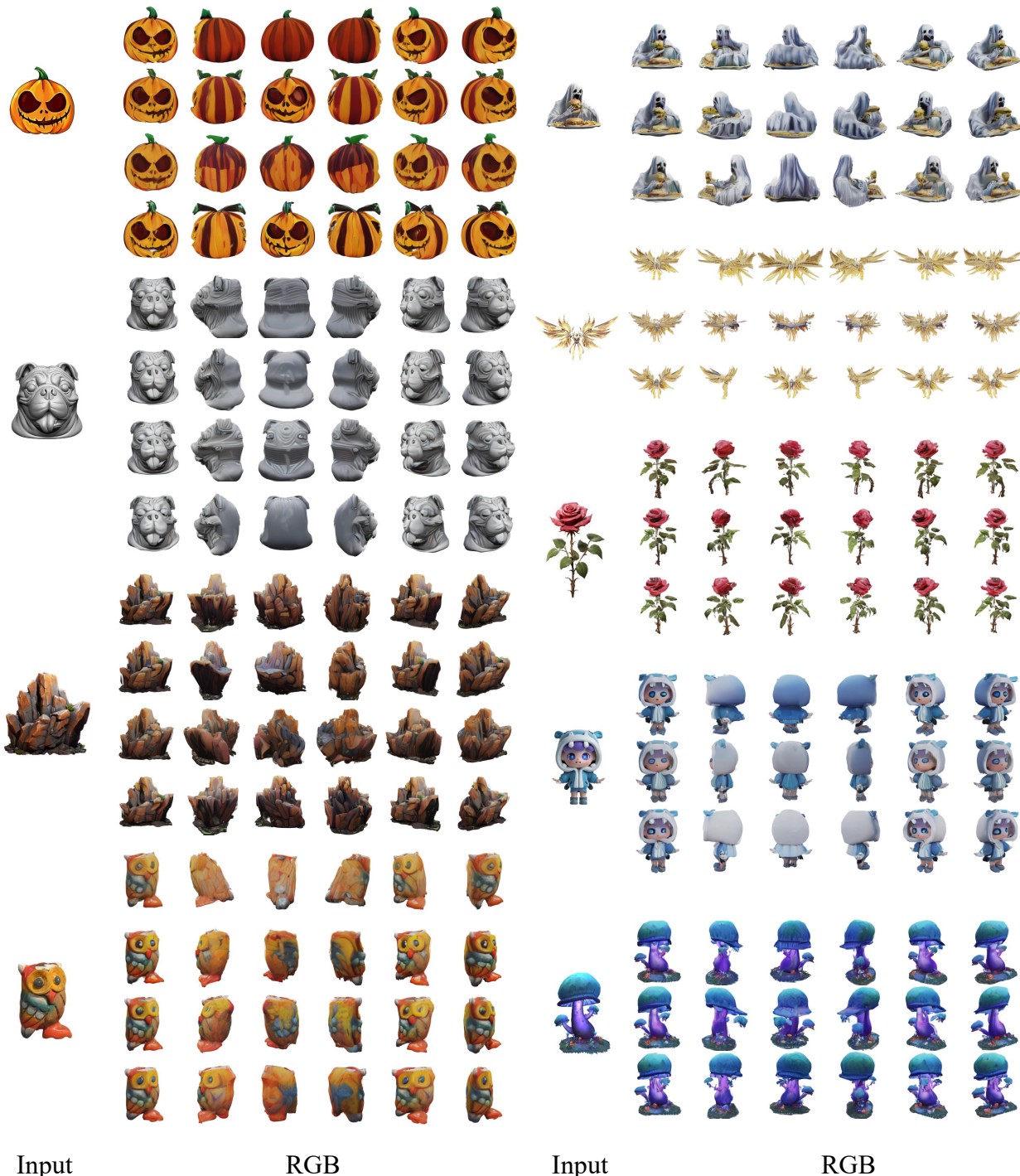

Input          RGB          Input          RGB

Figure 12: Generative 3D model with various geometry and texture given the same condition image, which shows the strong generative ability of our model.

