# OpenReview forum: "Zero-1-to-G: Taming Pretrained 2D Diffusion Model for Direct 3D Generation"
_TMLR — Accepted by TMLR_

### Review · Reviewer_KqUo · 2025-06-30

**Summary Of Contributions:**

This work proposes a Gaussian Splatting generator that generates the GS as a set of 2D images of different views and attributes (position, opacity, scale, rotation, RGB). A 2D diffusion model pretrained on natural images is fine-tuned to generate these decomposed 2D sets.

The original VAE encoder is reused to encode the set of 2D images, and the decoder is fine-tuned on the attribute images to reduce reconstruction artifacts. The 2D diffusion prior is initialized from a model pretrained on 2D natural images. The total training cost of the proposed method is lower than other train-from-scratch methods.

**Audience:**

Yes

**Claims And Evidence:**

Yes

**Requested Changes:**

- There are a few missing citations, for example:
  - 3DGen: Triplane Latent Diffusion for Textured Mesh Generation
     - there may be other related work in this direction which use 2D diffusion on 2D latent images of triplane.
  - CRM: Single Image to 3D Textured Mesh with Convolutional Reconstruction Model

- If the method turns out to be more robust than other regression-based methods on out-of-distribution data, it would make the story more convincing. In the Appendix, you show some results on real-world cases—are they better than other methods?
- What’s the resolution of the splatter images? Do you see that increasing the resolution leads to better quality?

**Strengths And Weaknesses:**

Pros:
- The proposed method utilizes a 2D diffusion prior pretrained on natural images without additional distillation steps.
- Compared to training a model from scratch, the training cost is much lower.
- Paper is well written and clear. The design of the model is reasonable, and ablation results support each design choice.

Cons
- Compared to regression methods, the inference speed is slower. But it might be possible to improve the speed through distillation and reducing the denoising steps.
- The quality of the generated shapes is not very impressive. Although in the paper it shows that it’s better than InstantMesh and LRM, it seems worse than other more recent methods that are not included or cited, for example, the Convolutional Reconstruction Model (CRM) work in ECCV 2024.

Question:
- Do you re-compute the mean and std for the latent space for your dataset before fine-tuning the latent diffusion model? Do you find it improves the generation quality?
- There are many works that fine-tuned Stable Diffusion into multiview RGB & normal generation. Do you think initializing the model weights from a multiview prior could improve the quality? On page 20, figure 12, right column, 4th row, the girl character wearing blue clothes—the generated front view does not look very good or consistent with the input view.

---

### Review · Reviewer_XWVG · 2025-07-06

**Summary Of Contributions:**

This paper proposes Zero-1-to-G, a framework that can directly generate 3D objects utilizing the generative priors learned in a 2D diffusion model.
Zero-1-to-G adopts splatter images, a representation that decomposes Gaussian splats into multi-view attribute images.
Then the proposed method finetunes a pretrained Stable Diffusion model to predict multi-view attribute images.
Cross-view and cross-attribute modules are designed and incorporated into the diffusion model to enhance its performance.
Experiments show that the proposed method outperforms existing image-to-3D methods both qualitatively and quantitatively.
The design decisions made in this paper are also supported by ablation studies.

**Audience:**

Yes

**Broader Impact Concerns:**

The paper mentions the concerns in the Broader Impact Statement section. I don't see outstanding concerns.

**Claims And Evidence:**

Yes

**Requested Changes:**

Please see the weaknesses mentioned above for the requested changes:

1. Please clarify how "fusion" is done and how that process enhances multi-view consistency.
2. I encourage the authors to add more discussion and comparison to the "concurrent works". I'm curious to see how different 2D representations of 3D (splatter images, UV maps, etc) have different effects.
3. Please reconsider whether the work can be claimed to be the "first" as mentioned above.

Minor comments:

1. "Compared to other direct 3D generation method LN3Diff (Lan et al., 2024), it struggles with fine-grained ..." It's not clear what "it" means here.

**Strengths And Weaknesses:**

Strengths:

1. The paper provides an insight that although pretrained 2D diffusion VAE produces visually-appealing 2D reconstruction results, when using the results for 3D reconstruction, noticeable artifacts appear. This observation motivates the proposed finetuning of the decoder in the VAE model, which is shown to be effective through experiments. This insight can be inspiring to the research community.
2. Zero-1-to-G effectively utilizes the prior learned in the 2D diffusion models and adapts them for 3D generation through splatter images.
3. Cross-view and cross-attribute attention modules are designed to enhance the generation performance.
4. Experiments show that the proposed method outperforms compared baselines, and ablation studies verify the effectiveness of the design decisions made in the method.
5. This paper is well-written and easy to follow.


Weaknesses:

1. It's not clear how the multi-view splatter images are "fused into Gaussian splats" (this is only mentioned once in Figure 2). There are probably overlapping reconstructed regions across different views. If the proposed method simply takes the union of Gaussians from all different splatter images, there are probably redundant Gaussians (those corresponding to the same 3D region but appearing on different splatter images), and they may cause multi-view inconsistencies. Without such detail, it is not clear how "eliminating the need for pixel-level correspondence between multiview splatter images" is achieved.
2. This paper claims to be "the first direct image-to-3D generative model to effectively utilize pretrained 2D diffusion priors". However, seems Omage and GIMDiffusion have already proposed such systems before. While this paper claims Omage and GIMDiffusion to be "concurrent work", they were released about 10-11 months ago at this point. I think they should not be considered "concurrent," and a necessary comparison is needed.

---

### Review · Reviewer_bBtn · 2025-07-10

**Summary Of Contributions:**

This paper focuses on the task of generating 3D objects from a single image. The authors propose to leverage a pre-trained 2D image generation model to predict multi-view axis-aligned 3D Gaussian distributions to ensure multi-view consistency. The authors perform a detailed ablation study and compare it with several baseline methods to demonstrate the effectiveness of the proposed method.

**Audience:**

Yes

**Claims And Evidence:**

Yes

**Requested Changes:**

Please see the above "Weaknesses" section.

**Strengths And Weaknesses:**

Strengths
- I like the idea of ​​repurposing the image 2D diffusion model for 3D generation. Several papers have demonstrated this, including: Kiss3dgen: Repurposing image diffusion models for 3d asset generation, CVPR2025.
- Overall, the solution is reasonable and technically feasible.

Weaknesses
- I think this paper is closely related to the already published paper "Direct and Explicit 3D Generation from a Single Image" (3DV 2025). The ideas proposed in the paper are roughly similar - re-purposing the 2D image diffusion model to predict axis-aligned 3D Gaussian maps, while the 3DV paper enforces 3D multi-view consistency through the epipolar attention mechanism on the decoder and a new novel synthesis loss. Although the code has not been released yet, I think the authors should clearly explain the differences and advantages of the 3DV'25 paper.


- I am actually a little concerned about the comparison with existing methods. The 3DV paper shows more comparisons with SoTA, as shown in Table 1. It would be better for the authors to perform a more thorough comparison to verify the effectiveness of the proposed method.

- It would be good to do a more detailed ablation study on the impact of the number of views and the tradeoff with performance and latency. My feeling is that the more views predicted, the better the results, especially for shapes with complex geometry.

---

### Decision · Action_Editor_Apdn · 2025-08-20

**Recommendation:** Accept with minor revision

**Additional Comments:**

Although the reviewers are satisfied with the revised paper which they feel has addressed their concerns, it should be noted that the experiments on higher-resolution images as shown in Table 4 of the revised paper show that more investigation needs to be done on that aspect. While it is fine with me to leave this aspect to future work, it would be good to extend the discussions there (and possibly also add some more experiments) to provide insights to inspire future investigation. This should be doable within the page limit of TMLR.

**Audience:**

Yes

**Audience Explanation:**

Yes, researchers interested in 3D generation in the TMLR's audience will find this paper very relevant.

**Claims And Evidence:**

Yes

**Claims Explanation:**

This paper presents an attempt to address the challenges encountered when using pre-trained 2D diffusion models for 3D reconstruction through 3D Gaussian splatting. In particular, it seeks to fine-tune the decoder appropriately to reduce the chance of obtaining artifacts and ensure multi-view consistency during reconstruction.

The paper has merits which include effectiveness of the proposed method in comparative studies and clarity of its presentation. That said, some issues on the original submission were raised by the reviewers, including some potentially related methods that should also be compared.

We thank the authors for considering the comments and suggestions of the reviewers seriously in their responses. The original paper has been revised by conducting more experiments involving more baselines and settings, including more references that are also related to the scope of the paper, and adding more discussions to address the questions and comments.

---

> ### Author Response · Authors · 2025-09-16
>
> We sincerely thank you for the additional suggestions. In the camera-ready version, we have made the following revisions:
> + We addressed all reviewer recommendations, including additional ablations on the number of splatter images and the resolution of each splatter image, expanded baseline comparisons with state-of-the-art methods, and added detailed explanations of failure cases in the appendix as well as references to more related works. We believe these improvements make our work more comprehensive and clarify how it stands in the related literature.
> + Regarding the minor revision suggestion, we have extended the discussion on generating higher-resolution splatter images for both training and inference. We think that fine-tuning from low-resolution splatter model and using more compuational resources should make the high-resolution model work. We have also outlined possible directions to address these challenges in future work.